# Consideration of SHP-1 as a Molecular Target for Tumor Therapy

**DOI:** 10.3390/ijms25010331

**Published:** 2023-12-26

**Authors:** Seyeon Lim, Ki Won Lee, Jeong Yoon Kim, Kwang Dong Kim

**Affiliations:** 1Division of Applied Life Science (BK21 Plus), Gyeongsang National University, Jinju 52828, Republic of Korea; yeon21@gnu.ac.kr; 2Anti-Aging Bio Cell Factory—Regional Leading Research Center, Gyeongsang National University, Jinju 52828, Republic of Korea; leemaskup@gnu.ac.kr; 3Department of Pharmaceutical Engineering, Institute of Agricultural and Life Science (IALS), Gyeongsang National University, Jinju 52725, Republic of Korea; jykim21@gnu.ac.kr; 4Plant Molecular Biology and Biotechnology Research Center (PMBBRC), Gyeongsang National University, Jinju 52828, Republic of Korea

**Keywords:** receptor tyrosine kinases, protein tyrosine phosphatases, Src homology region 2 (SH2) domain-containing phosphatase 1, anticancer drug

## Abstract

Abnormal activation of receptor tyrosine kinases (RTKs) contributes to tumorigenesis, while protein tyrosine phosphatases (PTPs) contribute to tumor control. One of the most representative PTPs is Src homology region 2 (SH2) domain-containing phosphatase 1 (SHP-1), which is associated with either an increased or decreased survival rate depending on the cancer type. Hypermethylation in the promoter region of *PTPN6*, the gene for the SHP-1 protein, is a representative epigenetic regulation mechanism that suppresses the expression of SHP-1 in tumor cells. SHP-1 comprises two SH2 domains (N-SH2 and C-SH2) and a catalytic PTP domain. Intramolecular interactions between the N-SH2 and PTP domains inhibit SHP-1 activity. Opening of the PTP domain by a conformational change in SHP-1 increases enzymatic activity and contributes to a tumor control phenotype by inhibiting the activation of the Janus kinase/signal transducer and activator of transcription (JAK/STAT3) pathway. Although various compounds that increase SHP-1 activation or expression have been proposed as tumor therapeutics, except sorafenib and its derivatives, few candidates have demonstrated clinical significance. In some cancers, SHP-1 expression and activation contribute to a tumorigenic phenotype by inducing a tumor-friendly microenvironment. Therefore, developing anticancer drugs targeting SHP-1 must consider the effect of SHP-1 on both cell biological mechanisms of SHP-1 in tumor cells and the tumor microenvironment according to the target cancer type. Furthermore, the use of combination therapies should be considered.

## 1. Introduction

Protein tyrosine kinases (RTKs), particularly receptor tyrosine kinases, are well known for their contribution to tumorigenesis. RTKs mediate intercellular communication, controlling a wide range of biological functions, including cell division, cell motility, and differentiation. Mutations responsible for abnormal activation of genes encoding RTKs, such as EGFR, HER2/ErbB2, and MET, have been identified in various cancer types [1]. Many studies have been conducted on how they contribute to cancer development, and the RTKs have been proposed as targets for developing tumor therapies [2,3]. RTKs are activated by receptor-specific ligands, such as growth factors, leading to receptor dimerization or oligomerization. Dimerization induces activation of the intracellular tyrosine kinase domain through trans-autophosphorylation.

In contrast to RTKs, phosphatases are enzymes that dephosphorylate substrate proteins, and among them, protein tyrosine phosphatase (PTP) is thought to inhibit RTK function by dephosphorylating proteins phosphorylated by RTK. However, not all PTPs inhibit PTK-mediated tumorigenesis, and PTPs that contribute to tumorigenesis were recently reported [4,5,6]. Among the PTPs, Src homology region 2 (SH2) domain-containing phosphatase 1 (SHP-1) is the first known SHP and is involved in cell cycle control, cancer cell migration and invasion, and apoptosis induction. In this review, we summarize the regulation of SHP-1 expression, the relationship between SHP-1 expression and tumors, the regulation of SHP-1-mediated cell signaling, the function of SHP-1 in the tumor microenvironment, and the use of SHP-1 in the development of tumor therapeutics.

## 2. Epigenetic Regulation of SHP-1 Expression

*PTPN6* encodes SHP-1, a nonreceptor tyrosine phosphatase. *PTPN6* mutations have been reported in several cancers; the total incidence of such mutations is 2.7% (uterine carcinosarcoma, 7.01%; testicular germ cell carcinoma, 6.04%; ovarian cancer, 5.82%; and melanoma, 5.18%). Phosphatase mutations occur most frequently (42.27%); however, there are few reports on their correlation with the SHP-1 function, which is scarce [7]. Two proteins of different sizes can be synthesized from different translation initiation codons located in exon 1 and of the *PTPN6* gene, located on human chromosome 12p13 [8,9]. The two proteins show similar enzymatic activity [8,10]. Promoter 1, located upstream of exon 1, is activated in epithelial-origin cells [11], and promoter 2, located upstream of exon 2, is activated in hematopoietic cells [12]. Promoter 1 is activated by various transcription factors, including NFκB, upstream stimulatory factor 1 (USF-1), or NF-Y [11,13]; promoter 2 can be activated by NFκB p65 and PU.1 in hematopoietic cells (Figure 1A) [14,15].

Epigenetic silencing affects SHP-1 expression in tumor cells. DNA methylation is a crucial epigenetic mechanism in regulating gene expression, with reports that hypomethylated DNA is associated with tumorigenesis and tumor development [16]. Reports suggest the transcriptional repression of the *PTNP6* promoter by hypermethylated CpG islands during the regulation of SHP-1 expression, which shows tumor suppressor activity in hematological malignancies [17,18,19], esophageal squamous cell carcinoma [20], and gastric adenocarcinoma [21,22]. To date, three types of DNA methyltransferases (DNMT) have been identified: DNMT1, DNMT2, and DNMT3a/b. [23]. Among them, DNMT1 controls SHP-1 expression by inducing aberrant methylation on promoter 2 of *PTPN6* in chronic myelogenous leukemia cells [24]. SHP-1 expression has an inverse relationship with DNMT1 and STAT3; its expression decreases when the activation of DNMT1 and STAT3 in tumor cells increases [25]. This is because activated STAT3 induces DNMT1 expression [26]. STAT3-DNMT1 interaction [27], which requires STAT3 acetylation [28], mediates DNMT1-mediated epigenetic gene silencing [28]. Therefore, the activated STAT3 inhibits SHP-1 expression via DNMT1 [27]. In carcinoma-associated fibroblasts (CAFs), which mediate the initiation of a pro-invasive tumor microenvironment, p300-histone acetyltransferase acetylates STAT3, which in turn upregulates and activates DNMT3b DNA methyltransferase. DNMT3b represses SHP-1 expression via CpG motif methylation of the *PTPN6* promoter. Consistently, in human lung and head and neck carcinoma, STAT3 acetylation and phosphorylation are inversely correlated with SHP-1 expression [29].

In addition to DNA methylation, histone acetylation is another representative epigenetic gene expression regulation mechanism. In a cohort of 37 patients with diffuse large B-cell lymphoma (DLBL), hypermethylation of the P2 promoter of *PTPN6* was only observed in 57% of patients. When treated with a DNA methyltransferase inhibitor (5-aza-deoxycytidine) and histone deacetylase (HDAC) inhibitor (LBH589), the inhibition of *PTPN6* expression in DLBL cells was reversed. LBH589 induces SHP-1 expression by increasing the H3K9Ac mark within the *PTPN6* P2 promoter [30]. Although LBH589 induces SHP-1 expression in chronic myeloid leukemia, HDAC does not directly combine with the *PTPN6* promoter [31]. HDAC3 and DNMT1 expression are increased in hypertrophy cell models and high-fat diet rat models. It was reported that HDAC3-mediated DNMT1 deacetylation causes an increase in DNMT1 stability. Therefore, an increase in DNMT1 suppresses SHP-1 expression (Figure 1B) [32]. Although promoter methylation is a relatively well-established epigenetic gene expression regulation mechanism of SHP-1, more precise mechanisms for HDAC-mediated regulation need to be investigated.

## 3. SHP-1 Expression in Tumors

SHP-1 expression may be regarded as a prognostic marker associated with decreased and increased tumor pathological symptoms. The association between SHP-1 expression and patient survival differs based on tumor type (Table 1). High levels of SHP-1 expression are related to high survival rates in patients with tumors, including breast cancer, esophageal squamous cell carcinoma, hepatocellular carcinoma, and prostate cancer [20,33,34,35,36,37]. Additionally, there is a report on the antitumor function of SHP-1 in gastric cancer, although it is not associated with SHP-1 expression [38]. The expression of colon cancer-associated transcript (*CCAT5*) was upregulated in ascites-derived gastric cancer cells, and increased expression of *CCAT5* was found to be associated with poor patient prognosis. *CCAT5* binds to the C-end domain of STAT3 and inhibits STAT3^Y705^ dephosphorylation mediated by SHP-1, thereby inducing STAT3 nuclear entry and metastatic activation, which promotes gastric cancer progression. In contrast, few reports suggest that SHP-1 expression and tumor patient survival are inversely correlated in patients with acute myeloid leukemia, colorectal cancer, and glioblastoma [39,40,41]. Therefore, although SHP-1 is generally regarded as a phosphatase with a tumor suppressor function, its role in tumor prognosis is probably dependent on the cancer type. Although the mechanism of STAT3 suppression by SHP-1 as a tumor suppressor mechanism is well-established, the SHP-1-mediated tumor-friendly molecular mechanism is not.

## 4. The Function of SHP-1 and Tumors

### 4.1. SHP-1 Structure and Its Activity

SHP-1 comprises two N-terminal SH2 domains (N-SH2 and C-SH2), a PTP domain, and a C-terminal tail containing several phosphorylation sites and a nuclear localization signal [42]. A structural study on SHP-1 lacking the C-terminal tail revealed that the intramolecular interaction of the N-SH2 domain with the PTP domain forms an autoinhibitory form of SHP-1, preventing the N-SH2 domain from exposing Cys455 of the active site and blocking substrate access to the active site. This autoinhibitory form is further stabilized by hydrogen bonding and salt bridge formation between the N-SH2 domain and PTP domain residues. The C-SH2 domain has been proposed as flexible and mobile and might play a role in sensing phosphopeptides, thereby weakening the autoinhibitory interaction between the SH2 and PTP domains (Figure 2A) [43].

Furthermore, a recent study elucidating the structure of the full-length and active forms of SHP-1 has shown that when the two SH2 domains bind to a ligand, the flexible C-SH2 domain rotates, causing the N-SH2 domain to rearrange and detach from the ligand’s active site. In addition, newly identified interactions between the N-SH2 and PTP domains and between the two SH2 domains further stabilized the open conformation of SHP-1 [44]. SHP-1 activity can also be regulated by phosphorylation, and three phosphorylation sites have been discovered to date: Tyr536, Tyr564, and Ser591. Tyr536 and Tyr564 are phosphorylated by Src family kinases, leading to increased SHP-1 activity. Tyr536 phosphorylation increases SHP-1 activity by inducing interaction with the N-SH2 domain and inhibiting interaction with the PTPase domain. On the other hand, Tyr564 phosphorylation indirectly increases PTPase activity by binding to the C-SH2 domain [45]. Additionally, upon stimulation by cellular activation signals, protein kinase C phosphorylates C-terminal Ser591 of SHP-1, thereby inhibiting its phosphatase activity (Figure 2B) [46,47]. Thus, additional biochemical and molecular biology studies are necessary to clarify how the C-terminal tail regulates SHP-1 activity and function and how it interacts with the other three domains.

### 4.2. Antitumor Activity of SHP-1

Tyrosine phosphorylation of proteins is a reversible posttranslational modification regulated by tyrosine kinases and PTPs. A common feature of cancer progression is the abnormal activation of tyrosine kinases due to an imbalance between phosphorylation and dephosphorylation. The Janus kinase (JAK)/signal transducer and activator of transcription (STAT3) pathway is a representative tyrosine phosphorylation-mediated oncogenic signaling pathway.

STAT3 forms a dimer by phosphorylating tyrosine residues (Tyr705), and the dimerized STAT3 moves to the nucleus, where it acts as a transcription factor [48,49,50,51]. Unregulated activation of STAT3 occurs in various cancers and contributes to tumorigenesis [52,53]. As upstream tyrosine kinases of STAT3, JAKs transmit the signals from receptors/ligand interactions to STAT3. Various ligands, such as IL-6, IFNs, PDGF, TGF, IGF, and EGF, activate STAT3 via their receptor-associated JAK activation [54,55,56,57,58,59,60]. The IL6/JAK2/STAT3 signaling axis is best studied in the context of tumor metastasis and tumor pathology since it induces epithelial–mesenchymal transition (EMT), resulting in tumor metastasis by promoting the expression of EMT-inducing transcription factors, such as Snail, ZEB1, JunB, and Twist1 [51,61,62,63,64]. Therefore, regulating the JAK/STAT3 signaling axis may be an effective target in tumor therapy. SHP-1 is a representative tyrosine phosphatase associated with tumor suppression and is a negative regulator of receptor-related signaling of three families: growth factor receptors with tyrosine kinase activity [65,66,67], cytokine receptors [60,68,69] and receptors involved in the immune response [70,71,72,73]. Generally, SHP-1-mediated inhibition of the JAK/STAT3 pathway is inversely correlated with tumor progression, aggressiveness, and metastasis [74,75,76]. SHP-1 binding to erythropoietin receptor (EPOR) inhibits erythropoietin (EPO)-mediated cell proliferation by inducing JAK2 dephosphorylation [69,77]. Furthermore, SHP-1 acts to dephosphorylate JAK1 in the IFN-α receptor signaling pathway [68]. SHP-1 silences the JAK/STAT pathway by inducing the dephosphorylation of both JAK and STAT3, and the loss of SHP-1 expression enhances JAK/STAT3 signaling in large cell lymphoma [78].

SHP-1 was reported to exert tyrosine phosphatase activity that directly downregulated pSTAT3 (Tyr705) and was a potent inhibitor of EMT in HCC and colorectal cancer (CRC) (Figure 3) [79,80]. Studies on several small molecules showing anticancer efficacy more clearly suggested SHP-1-mediated inhibition of the JAK/STAT3 signaling axis. Treatment with small molecules, such as 1′-acetoxychavicol acetate (ACA) [81], plumbagin [82], and allicin [83], significantly inhibited STAT3 through induction of SHP-1 in several types of cancer cells, including breast cancer, gastric cancer, and cholangiocarcinoma.

### 4.3. Association of SHP-1 with Tumorigenesis

The molecular mechanisms underlying the function of SHP-1 as a protein associated with tumorigenesis are not as well understood as those underlying the function of SHP-1 as a tumor suppressor. However, SHP-1 has been suggested to be associated with pro-tumorigenesis in some cancers. P53 inhibits SHP-1 expression, which reduces the Inhibition of SHP-1 expression by p53 reduces the proliferation of breast cancer cells by inducing trkA-Tyr674/Tyr675 phosphorylation [84]. Altered SHP-1 expression leads to changes in some components of the cell cycle. In ovarian cancer, where SHP-1 expression levels are high, inhibiting SHP-1 expression gradually reduces tumor growth by increasing the intracellular levels of Cdk2/p27 Kip1 and Cdk2/SHP-1 complex [85], which is opposite to the mechanism used by SHP-1 to inhibit cell division.

Additionally, SHP-1 deficiency in prostate cancer resulted in p27 accumulation, CDK6 reduction, retinoblastoma protein hypophosphorylation, cyclin E-CDK2 inhibition, and cycle arrest in phase G_1_ [86]. A common cause of radiotherapy failure in many tumors is radiation resistance, and the degree of radiosensitivity varies among tumor cells. SHP-1 has been found to reduce radiosensitization, and SHP-1 overexpression in the nasopharyngeal carcinoma cell line CNE-2 caused radiation resistance, which, in turn, reduced apoptosis by enhancing DNA double-strand break repair and increasing cell cycle arrest in phase S [87,88,89]. The molecular mechanisms underlying the double-edged sword of SHP-1’s effect on tumorigenesis remain poorly understood, suggesting differential protein expression pools in various cancers and differences in oncogenic signaling.

### 4.4. SHP-1-Related Small Molecules for Tumor Therapy

SHP-1 is a candidate molecular target for anticancer drug development because it regulates tumor growth and progression by downregulating JAK/STAT3 signaling. Natural products have recently received considerable attention regarding their potential as antitumor drugs, and several small molecules that inhibit STAT3 activity by inducing SHP-1 have been proposed as anticancer drug candidates (Table 2). To date, these small molecules have only been observed to have limited effects at the cellular level or in animal models. The Food and Drug Administration has approved two SHP-1-related anticancer drugs: sorafenib and regorafenib (Figure 4). Sorafenib is a multikinase inhibitor that promotes apoptosis by targeting STAT3 signaling in a variety of carcinomas, including pancreatic cancer and glioblastoma [90,91]. Sorafenib also increases the enzymatic activity of SHP-1 through direct interaction between the N-SH2 and the PTP domains in HCC cells, thereby downregulating STAT3 activity. Among the sorafenib derivatives, SC-43 and SC-40 were reported as more potent SHP-1 agonists than sorafenib and showed therapeutic potential for HCC treatment [92]. Additionally, SC-43 was confirmed to act as an SHP-1 agonist in cholangiocarcinoma [93], CRC [94], and breast cancer [95]. SC-60, another sorafenib derivative, also had an anticancer effect by increasing SHP-1 activity in HCC and triple-negative breast cancer (TNBC) [93,96,97]. Regorafenib is a multiple protein kinases inhibitor that is very similar to sorafenib, and it enhances SHP-1 activity in HCC and CRC to promote apoptosis by inhibiting STAT3 signaling [98,99]. SC-78, a derivative of regorafenib, also inhibits tumor growth and metastasis in TNBC by interfering with the paracrine and autocrine pathways of VEGF-A through the SHP-1/STAT3 signaling axis [100].

Although SHP-1 is generally known as a tumor suppressor, its expression is upregulated in some high-grade breast cancers [101] and ovarian cancers [102]. Substantial inhibitors of SHP-1 phosphatase activity have been developed and are undergoing preclinical and clinical studies at present, including NSC-87877, sodium stibogluconate (SSG), tyrosine phosphatase inhibitor 1, and suramin, but only a few have shown antitumor activity in experimental tumor models [93]. SSG has undergone Phase I trials for both malignant melanoma (NCT00498979) and advanced malignancies (NCT00629200), but no significant effect on tumor development was reported [103,104].

**Table 2 ijms-25-00331-t002:** Natural compound-mediated STAT3 inhibition through SHP-1 induction.

Compound	Source	Cancer Type	Remarks	Ref
Guggulsterone	*Commiphora mukul*	Myeloma	Guggulsterone suppressed the expression of STAT3-associated antiapoptotic gene products and enhanced the anticancer effects of bortezomib.	[105,106]
Morin	*Moraceae* family	Myeloma	The number and position of hydroxyl groups in the B ring of flavonols are important inhibitors of STAT3 activation.	[107]
Genipin	*Gardenia*	Myeloma	Genipin effectively enhanced the cytotoxic effects of anticancer drugs such as bortezomib, thalidomide, and paclitaxel.	[108]
Capillarisin (CPS)	*Artemisia capillaries*	Myeloma	CPS induced cell cycle arrest in the sub-G1 phase and enhanced the anticancer effects of bortezomib.	[109]
Ergosterol peroxide (EP)	*Ganoderma lucidum*	Myeloma	EP inhibited the growth of U266 cells inoculated into female BALB/c mice and effectively reduced STAT3 activity and CD34 expression.	[110]
Icariside II	*Epimedium koreanum*	Myeloid leukemia	SHP-1 inhibition using siRNA significantly blocked icariside II-induced STAT3 inactivation and apoptosis in U937 cells.	[111]
Honokiol (HNK)	*Magnolia officinalis*	Myeloid leukemia	HNK induces the expression of SHP-1 by increasing the expression of its related transcription factor, PU.1.	[112]
Zerumbone	*Zingiber zerumbet*	Renal cell carcinoma	Zerumbone inhibited growth of human RCC xenograft tumors and STAT3 activation in athymic nu/nu mice.	[113,114]
α-mangostin (α-MGT)	*Mangosteen*	Hepatocellular carcinoma	α-MGT exhibited anti-HCC effects by inhibiting SHP-1 degradation induced by the ubiquitin–proteasome pathway.	[115]
Emodin	*Rheum palmatum*	Hepato cellular carcinoma (HCC)	Emodin inhibited human HCC orthotopic tumor growth and STAT3 activation in athymic male nu/nu mice.	[116]
Plumbagin	*Plumbago zeylanica*	Gastric cancer	Plumbagin not only induced apoptosis but also inhibited gastric cancer cell proliferation, migration, and invasion.	[82]
Allicin	*Garlic*	Cholangiocarcinoma(CCA)	Allicin inhibited CCA cell migration, invasion, and EMT and induced cell death. It also attenuated CCA tumor growth in a nude mouse model.	[83]
1′-acetoxychavicol acetate (ACA)	*Languas galanga*	Breast cancer	ACA potently inhibited osteolysis in a mouse breast cancer skeletal metastasis model through the SHP-1/STAT3/MMPs signaling pathway.	[81]
Pectolinarigenin	*Cirsium chanroenicum*	Osteosarcoma	Pectolinarigenin interfered with the STAT3/DNMT1/HDAC1 complex formation at the SHP-1 promoter.	[117]

### 4.5. The Function of SHP-1 and SHP-2 in the Tumor Microenvironment

SHP-1 affects not only tumor cells but also the tumor microenvironment. From the perspective of tumor treatment, the antitumor effect of SHP-1 on tumor cells is expected, while in the tumor immune environment, the function of SHP-1 is generally tumor-friendly. SHP-1 is expressed in all mature hematopoietic lineages. Notably, its regulatory mechanism related to T-cell activation is thought to be closely related to tumor targeting activity and tumor treatment in the tumor microenvironment. SHP-1 limits antigen-specific T-cell activation by dephosphorylating the T-cell receptor (TCR) ζ chain or downstream adapter proteins, such as lymphocyte-cell-specific protein-tyrosine kinase (Lck), ZAP70, Vav family proteins, and PI3K [42,118]. CD8^+^ T-cells from SHP-1 deficient motheaten mice [119] showed more stable and longer-lasting immunological synapses with antigen-presenting cells (APC), which reduced the T-cell activation threshold, thereby increasing the activation of T cells with low antigen specificity, leading to effective tumor suppression through tumor-specific effector T cells [120,121]. Knockout of SHP-1 in CD133 chimeric antigen receptor (CAR) T-cells significantly enhanced the cytolytic effect on CD133+ glioma cell lines by CAR T-cells and increased secretion of TNF-α, IL-2 and IFN-γ [122]. CAR T-cells currently approved by the US FDA possess a TCR-derived ζ chain as an intracellular activation domain in addition to a co-stimulatory (4-1BB or CD28) domain [123]. CARs containing CD3δ, CD3ε, or CD3γ cytoplasmic tails have outperformed conventional ζ CAR T-cells in vivo. Making CARs mutated to phenylalanine on the intracellular domain N-terminal tyrosine of CD3γ and CD3δ will only phosphorylate at the C-terminal tyrosine (BBγFY and BBδFY), and SHP-1 will preferentially bind to CARs containing that single phosphorylated BBδ, fine-tuning and balancing T-cell activation to prevent exhaustion and dysfunction [124]. SHP-1 is strongly activated on CD8^+^ nonlytic tumor-infiltrating lymphocytes (TILs), co-localizes with Lck on nonlytic TILs, and inhibiting SHP-1 on lytic TILs overcomes the inhibition of TIL cytolysis by tumors. Contact between nonlytic TILs and tumor cells activates SHP-1, which rapidly dephosphorylates the Lck activation motif (Tyr394), thereby inhibiting effector phase function [125].

Immune checkpoints suppress autoimmunity and contribute to maintaining immune homeostasis by limiting T-cell activation. The exhausted T cells express inhibitory receptors such as PD-1 as immune checkpoint molecules. Some tumor cells use immune checkpoints to acquire immune tolerance to tumor-specific T cells [126]. Strategies blocking the interaction between programmed cell death protein 1 (PD-1) and PD ligand (PD-L)1/2, or between cytotoxic T-lymphocyte associated protein 4 (CTLA-4) and CD80/CD86 have been developed to overcome the immune checkpoints in tumor tissue. SHP-1 is recruited to the cytoplasmic tail of PD-1 by binding to immunoreceptor tyrosine-based switch motifs (ITSM), which then induces dephosphorylation and inactivation of proximal signaling molecules activated through TCRs [127,128]. Lymphocyte-activating gene 3 (LAG3) associates with SHP-1/2, and LAG3/PD-1 collaboration limits CD8^+^ T-cell signaling, which dampened antitumor immunity in a murine ovarian cancer model [129]. Recently, it has also been reported that PD-1 interacts more effectively with SHP-2 than SHP-1. B and T-lymphocyte attenuator (BTLA), on the other hand, has been reported to interact more effectively with SHP-1 than SHP-2 due to an interaction between the immunoreceptor tyrosine-based inhibitory motif (ITIM) and the N-terminus of SHP-1 [130,131]. PD-1 interacts primarily with SHP-2 but also with SHP-1 in the absence of SHP-2, and both PD-1-SHP-1 and PD-1-SHP-2 complexes attenuate TCR and CD28 signaling pathways [132]. Although the interaction between SHP-1/SHP-2 and PD-1 is thought to contribute to T cell exhaustion, the *CD4cre Ptpn6/11*^fl/fl^ mice do not improve T cell-mediated tumor control. Depleting these phosphatases from the polyclonal T-cell compartment does not improve tumor control, suggesting that caution should be taken when considering their inhibition for immunotherapeutic approaches [133].

The function of regulatory T-cells (Tregs) in immune tolerance in the tumor microenvironment has been well studied, but the function of SHP-1 in the tumor microenvironment in relation to Tregs is paradoxical. Loss of SHP-1, a negative regulator of TCR signaling, renders naïve CD4^+^ and CD8^+^ T-cells resistant to Treg-mediated suppression in a T-cell-specific manner (Figure 5) [134]. On the other hand, loss of SHP-1 expression in Tregs significantly increases their capacity, and specific pharmacological inhibition of SHP-1 enzymatic activity via the anticancer drug SSG considerably increased Treg suppressive activity both in vivo and ex vivo [135]. Therefore, SHP-1 function differs according to the cell population, and the ability to control SHP-1 expression or function in different cell populations would be advantageous for tumor control.

## 5. Conclusions

With some exceptions, SHP-1-mediated inhibition of RTK signaling in cancer cells is generally associated with antitumor effects, and small molecules that increase SHP-1 expression and induce its activity may still be important candidate anticancer therapeutics. However, the clinical development of drugs that increase SHP-1 function or expression is extremely limited. Even if a drug candidate exerts a tumor-suppressing function specific to tumor cells, it is difficult to judge its overall effectiveness because it also suppresses the immune environment within the tumor tissue. In particular, it inhibits the activity of cytotoxic T lymphocytes, which display specific cytotoxicity against tumor cells. Therefore, when developing a tumor treatment targeting SHP-1, the function of SHP-1 in the type of cancer to be treated must first be determined, and the development of an effective combination treatment must also be considered. For example, for carcinomas with antitumor effects mediated by SHP-1, the combination of an SHP-1 agonist with SHP-1 knockdown or the CD3-mutated [124] CAR T-cell therapy may be considered.

## Figures and Tables

**Figure 1 ijms-25-00331-f001:**
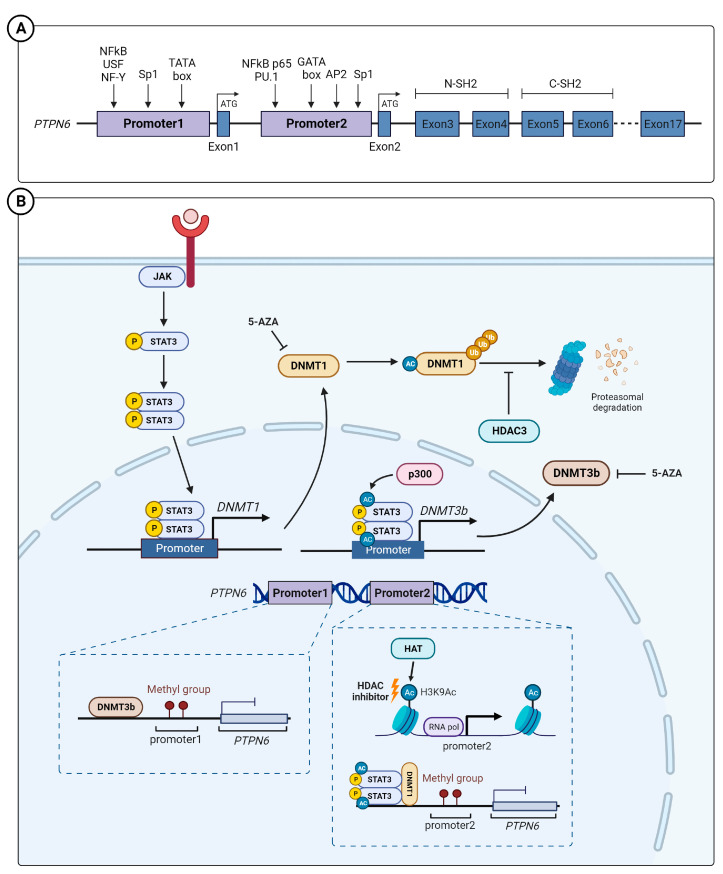
*PTPN6* gene structure and regulation of SHP-1 expression. (**A**) Schematic diagram of the SHP-1 gene (*PTPN6*). The *PTPN6* has 17 exons; promoters 1 and promoter 2 are located in exon 1 and exon 2, respectively. Promoter 1 and 2 are mainly active in epithelial and hematopoietic cells, respectively. (**B**) The epigenetic regulatory mechanism regulating SHP-1 expression. JAK induces phosphorylation of STAT3 (pSTAT3), and the pSTAT3 forms a dimer and induces the transcription of *DNMT3B*.The methylation of the SHP-1 promoter region via DNMT and H3L9 acetylation (Ac) via histone acetyltransferases can down- and upregulate SHP-1 expression, respectively. USF—upstream stimulatory factor; DNMT—DNA methyltransferases; HDAC—histone deacetylase; 5-AZA—5-Aza-2′-deoxycytidine.

**Figure 2 ijms-25-00331-f002:**
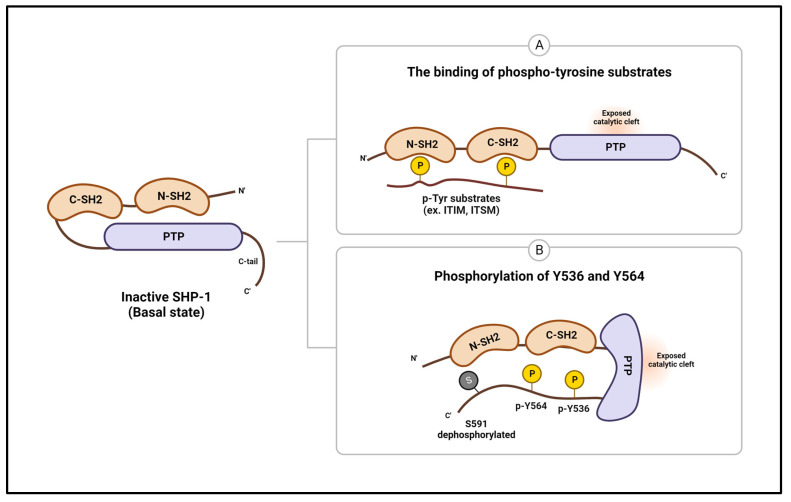
Protein structures and regulation of SHP-1. The enzymatic activity of SHP-1 is inhibited by intramolecular interaction between the N-SH2 and PTP domains. (**A**) The binding of the SH2 domain by tyrosine-phosphorylated substrates and (**B**) phosphorylation of tyrosine residues in the C-terminal tail causes a conformational change that opens the phosphatase active site and contributes to phosphatase activation. ITIM—immunoreceptor tyrosine-based inhibitory motif; ITSM—immunoreceptor tyrosine-based switch motif.

**Figure 3 ijms-25-00331-f003:**
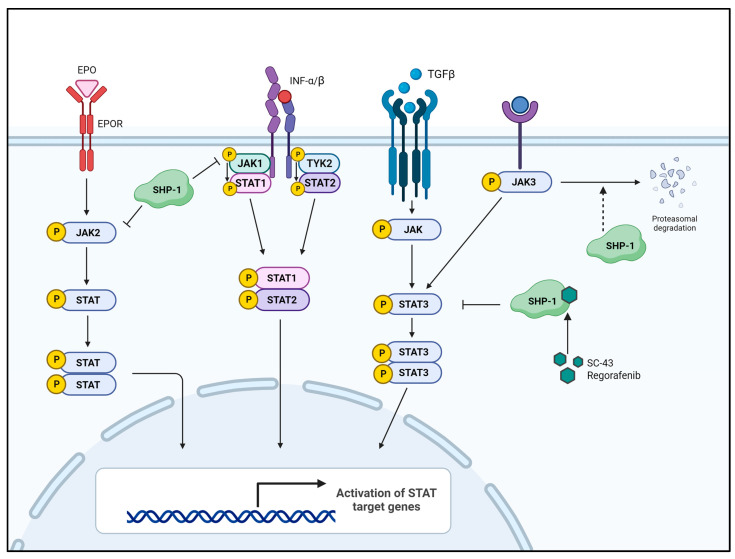
SHP-1-mediated inhibition of the JAK/STAT signaling pathway. Several growth factors and cytokines activate their associated receptors, which, in turn, activate JAK. Activated JAK then activates STAT through phosphorylation and moves the activated STAT (p-STAT) to the nucleus, upregulating the expression of STAT-related genes. SHP-1 directly dephosphorylates STAT3 or its upstream JAKs, thereby inhibiting cell proliferation, survival, migration, and invasion. EPO—erythropoietin; EPOR—erythropoietin receptor; TYK2—tyrosine kinase 2.

**Figure 4 ijms-25-00331-f004:**
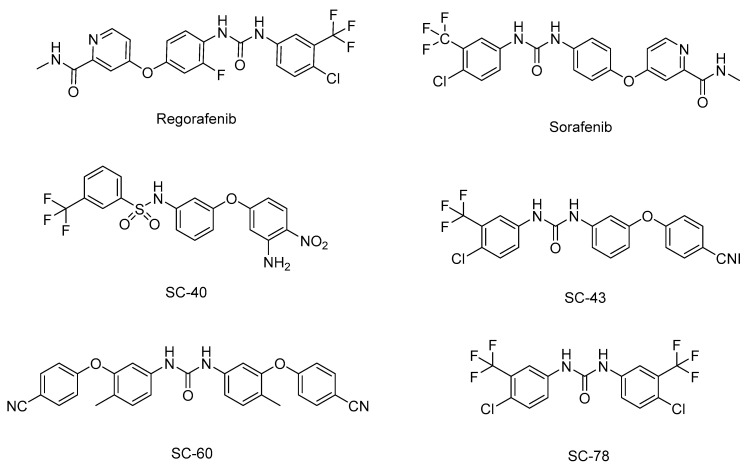
Chemical structures of SHP-1 agonists, regorafenib, and sorafenib and their derivatives.

**Figure 5 ijms-25-00331-f005:**
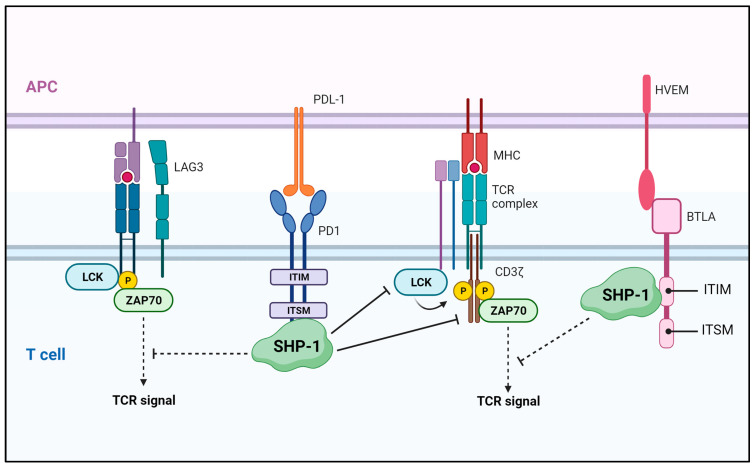
SHP-1-mediated regulation in the tumor microenvironment. SHP-1 blocks T cell activation by negatively regulating TCR signaling through binding to the ITSM site and ITIM domain of the coinhibitory molecules, PD-1 and BTLA, respectively. APC—antigen-presenting cell; LAG—lymphocyte-activating gene 3; LCK—lymphocyte-cell-specific protein-tyrosine kinase; ZAP70—zeta-chain-associated protein kinase 70; PD1—programmed cell death protein 1; HVEM—herpes virus entry mediator; BTLA—B and T-lymphocyte attenuator.

**Table 1 ijms-25-00331-t001:** Clinicopathological role of SHP-1 in tumors.

Tumor Type	Reported Findings	Predicted Function	Refs
Acute myeloid leukemia	SHP-1 expression was negatively correlated with the overall survival of leukemia patients.	Oncogenic	[39]
Breast cancer	SHP-1 expression is inversely correlated with pSTAT3 and positively correlated with recurrence-free survival in patients.	Tumor suppressive	[33]
	According to the TCGA database, high expression of SHP-1 was associated with better overall survival.	Tumor suppressive	[34]
	Higher SHP-1 expression associated with better overall survival.	Tumor suppressive	[35]
Colorectal cancer	The survival time of patients with high SHP-1 expression is shorter than those of patients with low SHP-1 expression.	Oncogenic	[40]
Esophageal squamous cell carcinoma	Negative correlation with the tumor-node metastasis staging system, pathological differentiation, and lymph node metastasis: The downregulation and hypermethylation of SHP-1 are associated with poor survival.	Tumor suppressive	[20]
Glioblastoma	Upregulation of SHP-1 in GBM patients according to TCGA analysis. High expression of SHP-1 was associated with advanced grade and poor overall survival of glioma.	Oncogenic	[41]
Hepatocellular carcinoma	Downregulation of SHP-1 in hepatocellular carcinoma (HCC) is negatively correlated with tumor growth and overall survival in patients with HCC and hepatitis B virus infection.	Tumor suppressive	[36]
Prostate cancer	A decreased SHP-1 expression is associated with higher proliferation rates and increased risk of recurrence or progression-free survival after radical prostatectomy for localized prostate cancer.	Tumor suppressive	[37]

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
