# Peer review of "Consideration of SHP-1 as a Molecular Target for Tumor Therapy"

_ijms, 2023, doi:10.3390/ijms25010331_

Round 1

Reviewer 1 Report

Comments and Suggestions for Authors

A review by Lim et al focuses on SHP-1 as a potential anti-cancer target. The manuscript is well written in general, and visualized by interesting figures.

1. Gene names should be consequently written in italics.

2. Table 1 - 'reported findings' are too descriptive, and can be written more concisely without a loss of information.

3. Table 1 - why these tumors only are included? For example, there are papers in melanoma studying the role of SHP-1, too.

4. Abbreviations used in the figures should be defined.

5. Conclusions/future perspectives should be put into more current knowledge. Only a minority of references were published after 2020.

Author Response

Thank you for reviewing our manuscript and providing suggestions to improve its quality. Based on your advice, I made the following modifications:

Q 1. Gene names should be consequently written in italics.

Thank you for your point. As you pointed out, we have replaced all gene names with italics.

PTPN6 in line21, CCAT5 in line 121.

Q2. Table 1 - 'reported findings' are too descriptive, and can be written more concisely without a loss of information

- Thank you for your kindness. Following your advice, we've revised Table 1 to be more concise.

Q3. Table 1 - why these tumors only are included? For example, there are papers in melanoma studying the role of SHP-1, too.

Following your advice, we carefully attempted to search for clinicopathological research papers related to SHP-1 in melanoma patients, but were unable to find appropriate references; however, a clinical trial in human melanoma patients is described with references. lines 258-260.

Q4.  Abbreviations used in the figures should be defined.

Thank you for your important point. We defined the abbreviations in all figures.

Q5. Conclusions/future perspectives should be put into more current knowledge. Only a minority of references were published after 2020.

Your point is very important to improve the quality of the information provided to the readers of this paper and we thank you. In our research, we realized that there are more recent reports on SHPs in the immune environment than on their intrinsic functions, so we have expanded the discussion of these studies and added a few more recent papers to the references.

Line 119~125: Ref 33, In Table 2: Ref 114, Line 280~290: Ref 122, 123, and 124, Line 311~318: Ref 132 and 133.

Reviewer 2 Report

Comments and Suggestions for Authors

The review article addressed SHP-1 as a possible target for tumour therapy. The article is well written; however, I would suggest changing the title since there is also relevant information about SHP-2.  I would also suggest to add information on CD8 see doi: 10.4049/jimmunol.2300462

and also some agonist structures as agonist of SHP-1 doi: 10.1016/j.cbi.2023.110780

After those minor changes, the article should be published.

Comments on the Quality of English Language

Minor grammatical mistakes

Author Response

Thank you for reviewing our manuscript and providing suggestions to improve its quality. Based on your advice, I made the following modifications:

Q 1. Changing the title since there is also relevant information about SHP-2

As you point out, there is also some information about SHP-2. Since the information in SHP-2 is limited to section 4.5, we decided it was more appropriate to change the title of section 4.5 than to change the title of the manuscript. We have changed the title to “4.5. The function of SHP-1 and SHP-2 in the tumor microenvironment”. We hope you understand our decision.

Q2. I would also suggest to add information on CD8

- I think your suggestions are very informative and improve the quality of our manuscript, and we added the information in line 280~290.

Q3. some agonist structures as agonist of SHP-1.

Following your advice, we have added the chemical structures of SHP-1 agonists described in the text, drawn as shown in Figure 4.